# Feasibility of Magnetic Resonance Cholangiopancreatography in Dogs—A Post-Mortem Study

**DOI:** 10.3390/ani13152517

**Published:** 2023-08-04

**Authors:** Vahideh Rahmani, Juha Peltonen, Shyrley Paola Amarilla, Dmitri Hmelnikov, Mirja Ruohoniemi, Thomas Spillmann

**Affiliations:** 1Department of Equine and Small Animal Medicine, Faculty of Veterinary Medicine, University of Helsinki, 00100 Helsinki, Finland; mirja.ruohoniemi@helsinki.fi (M.R.); thomas.spillmann@helsinki.fi (T.S.); 2HUS Medical Imaging Center, Helsinki University Hospital, University of Helsinki, 00100 Helsinki, Finland; juha.peltonen@hus.fi (J.P.); dmitri.hmelnikov@hus.fi (D.H.); 3Department of Veterinary Biosciences, Faculty of Veterinary Medicine, University of Helsinki, 00100 Helsinki, Finland; pamarilla@vet.una.py; 4Department of Pathological Sciences, Faculty of Veterinary Sciences, National University of Asuncion, San Lorenzo 111421, Paraguay

**Keywords:** biliary tract, pancreas, diagnostic imaging, magnetic resonance cholangiopancreatography

## Abstract

**Simple Summary:**

Diseases of the biliary tract and pancreas are common in dogs and occasionally challenging to diagnose accurately. The aim of our study was to examine the feasibility of a modern diagnostic imaging technique, magnetic resonance cholangiopancreatography, in cadavers of eight adult dogs. This noninvasive imaging technique is commonly used in humans to visualize the biliary tract and pancreatic ducts and has been found to be feasible also in cats, but its utility in dogs has not been reported to date. We assessed the visibility and measured the diameter of various pancreatobiliary ductal structures using this novel technique, and two other methods were used for comparison. Our study showed that this magnetic resonance imaging technique allows for reliable visualization of the biliary tract. However, fine structures, such as pancreatic ducts with diameters of less than 1 mm, were challenging to assess using our 1.5 Tesla machine. Further studies are warranted to adapt magnetic resonance cholangiopancreatography to the diagnostic needs of live dogs.

**Abstract:**

Magnetic resonance cholangiopancreatography (MRCP) is commonly used in humans and is also feasible in cats. The aim of this post-mortem study was to investigate the feasibility of MRCP in eight adult dogs by comparing the visibility and measured diameters of the biliary tract and pancreatic ducts in MRCP with those of fluoroscopic retrograde cholangiopancreatography (FRCP) and corrosion casting. In autopsy, six dogs had no evidence of hepatobiliary disorders, one had pancreatic pathology, and one had biliary pathology. The gallbladder (GB), cystic duct, and common bile duct (CBD) were visible in the MRCP images of all eight dogs. However, the extrahepatic ducts and pancreatic ducts were only variably visible. There was statistical agreement between MRCP and FRCP in measuring the diameters of the GB (fundus and body) and CBD (at papilla and extrahepatic ducts’ junction). The diameter measurements correlated between MRCP and corrosion casting. Our study showed that MRCP is feasible in dogs and allowed for proper visualization of the biliary tract. However, ducts with diameters of <1 mm were difficult to visualize using a 1.5 Tesla MRI machine. Further studies are warranted to apply MRCP in the diagnostic imaging of live dogs.

## 1. Introduction

Chronic diseases of the biliary tract and the pancreas have become increasingly identified pathologies in dogs [1,2]. These diseases, including gallbladder (GB) mucocele, cholecystitis, cholelithiasis, and extrahepatic biliary tract neoplasia [3], can result in substantial morbidity and mortality, which may be reduced by early recognition [4]. Although chronic pancreatitis is a common and clinically significant disease in dogs, it is still challenging to diagnose definitively and to differentiate from pancreatic neoplasia due to nonspecific clinical signs and laboratory findings, as well as the common clinical practice of rarely taking pancreatic biopsies [2,5,6].

There has been considerable progress in the diagnostic imaging of pancreatobiliary diseases in recent years. In human medicine, magnetic resonance cholangiopancreatography (MRCP) has become a standard component of noninvasive magnetic resonance imaging (MRI) to visualize the biliary tree and pancreatic ducts by producing a recognizable projectional image format of these structures in their native configuration [7,8,9]. Continued technical improvements and the use of hormonal stimulation of the exocrine pancreas with secretin have permitted a more accurate display of the ductal anatomy and expanded the clinical applications of MRCP, opening new areas for clinical research [10]. In veterinary medicine, the potential of MRCP has been shown with and without secretin stimulation in healthy cats and cats with cholangitis and pancreatitis [11,12]. In a recently published postmortem investigation on cats using a 1.5 Tesla MRI machine, MRCP was shown to be a feasible modality to visualize the biliary and pancreatic duct structures of diameters > 1 mm and to evaluate and determine the diameters of extrahepatic biliary structures, such as the GB, cystic duct, and CBD [13].

Endoscopic retrograde cholangiopancreatography (ERCP) is an established method in humans for the diagnosis and treatment of biliary tract diseases, including obstruction and chronic pancreatitis [14,15]. ERCP applies a combination of endoscopy and fluoroscopy to image the biliary tree and pancreatic ducts. Few studies in veterinary medicine have been performed using this technique in normal dogs [16] and in dogs with gastrointestinal diseases [17,18,19]. The diameter of the CBD has been measured by ERCP in both healthy and diseased dogs [16,17,19].

Corrosion casting is an established technique in anatomy to expose the void of any injectable duct, allowing detailed morphologic visualization of various ductal and pocketing structures [20]. An experimental study showed that corrosion casting is a feasible method for the visualization and quantitative assessment of anatomical details of the fine biliary tree in mice with obstructive cholestasis [21]. In another study in rats, microcorrosion casting allowed for evaluation of the intrahepatic biliary tree [22]. In a recent postmortem study on cats, corrosion casting using vinyl polysiloxane successfully revealed the ductal anatomy of the biliary tract and pancreatic duct [13].

To the best of our knowledge, there have been no previous studies focused on MRCP assessment of the biliary tract and pancreatic ducts in dogs. The main aim of the present study was to investigate the feasibility of MRCP in visualizing the biliary tract and pancreatic ducts in adult dogs with and without disorders of these structures. Another aim was to explore whether the diameters for bile ducts, GB, and pancreatic ducts measured in MRCP agree with those of fluoroscopic retrograde cholangiopancreatography (FRCP, a modified ERCP technique for cadavers) and correlate with those of corrosion casting.

We hypothesized that MRCP would provide reliable visualization of the biliary tract and pancreatic ducts as well as measurement of the ductal diameters in adult dogs. Additionally, we hypothesized that the ductal diameters measured in MRCP would agree with those determined in FRCP and would correlate with those of corrosion casting.

## 2. Materials and Methods

### 2.1. Study Design

This was a prospective, observational, analytical, proof-of-concept study performed on eight bodies of client-owned dogs donated to research after euthanasia at the Small Animal Teaching Hospital, University of Helsinki, Finland from April 2020 to April 2021. All dogs presented to the hospital only for euthanasia due to different ethically justifiable reasons. No diagnostic or treatment procedures were performed prior to euthanasia. The study protocol was approved by the University of Helsinki Viikki Campus Research Ethics Committee (Ethics statement 17/2021). Information on the dogs, including their weight, age, gender, and breed, were recorded. To minimize post-mortem changes, all bodies were kept in the refrigerator (2–4 °C) prior to investigation. Within 24 h after euthanasia, all dogs underwent successful MRCP and FRCP procedures, and they were then examined in autopsy and with corrosion casting.

### 2.2. Procedures

#### 2.2.1. MRCP

MRI scanning was performed using a human head coil and a 1.5 Tesla MRI scanner (Ingenia, Philips, The Netherlands). The dogs were positioned in dorsal recumbency, and all pulse sequences were localized and limited to the cranial aspect of the abdomen, from the cranial margin of the diaphragm to the caudal margin of the pancreas. The imaging protocol included seven sequences (Table 1). Three-dimensional turbo spin echo MRCP (3D-TSE-MRCP) images were saved in 2D using JiveX software (JiveX Review Client 4.4.5, Visus Technology Transfer GmbH, Bochum, Germany) and used for further diameter measurements.

#### 2.2.2. FRCP

Shortly after the MRI scan, FRCP was performed with the dog in dorsal recumbency. The abdominal midline was incised to expose the duodenum. An incision was made in the duodenal wall to detect the major and minor papillae. The pancreatic duct was cannulated using a feline urethral catheter (3Fr × 5.2 in, Buster Cat Catheter, side holes, KRUUSE, Langeskov, Denmark), and the CBD was cannulated using a standard ERCP catheter (Microinvasive-4341, Boston Scientific, Marlborough, MA, USA). Iodine contrast medium (Iomeprol 300, Bracco-Byk Gulden, Konstanz, Germany) was administered with manually controlled pressure to firstly fill the pancreatic ducts but avoid pancreatic parenchyma filling and secondly to fill the extrahepatic biliary tract (extrahepatic ducts, GB, cystic duct, and CDB) under fluoroscopic control (BV Libra, Philips, Amsterdam, The Netherlands). Injection continued until the contrast medium filled the GB or the proximal part of the pancreatic ducts became clearly visible. To visualize various parts of the biliary tract and pancreatic ducts without overlapping, several fluoroscopic images were taken. During imaging, a radiopaque ruler was placed under the animal for further calibration. FRCP images were saved using an image recorder (MediCapture, MediCap USB 200, Plymouth Meeting, PA, USA).

#### 2.2.3. Gross Pathology, Histopathology, and Corrosion Casting

Following MRCP and FRCP, the dog cadavers were examined at the Department of Veterinary Biosciences, Pathology Unit, Faculty of Veterinary Medicine, University of Helsinki. The examinations included routine autopsies, taking tissue samples for histopathology, and corrosion casting.

Gross pathology with special emphasis on the liver and pancreas was conducted based on the criteria of the World Small Animal Associations’ (WSAVA) Liver Standardization Group and the nomenclature of the European College of Veterinary Pathologists. For histopathological examinations, tissue samples of all liver lobes, the proximal and middle portions of the intrahepatic ducts, the body of the pancreas, and both pancreatic lobes were obtained. Each tissue sample for histopathology was fixed in 10% neutral buffered formalin and subjected to hematoxylin and eosin (H&E) staining. In cases of suspected pancreatic fibrosis, the pancreas was stained with Masson’s trichrome stains for confirmation. Corrosion casting was conducted by manually controlled injection of vinyl polysiloxane (VPS; ExpressTM 2 VPS Impression Materials; 3M ESPE AG; Seefeld, Germany) into the extrahepatic and pancreatic ducts through the major and minor duodenal papillae, respectively. The liver and pancreas were digested using 30% sodium hydroxide solution (NaOH, sodium hydroxide ACS reagent, ≥97.0%, Pelletsun, Sigma-Aldrich, Darmstadt, Germany) to obtain the negative polymer impression for structural comparison and diameter measurements. Pictures of corrosion casts were taken with a Nikon digital camera (D70, Phra Nakhon Si Ayutthaya, Thailand) and were thereafter used for diameter measurements. A ruler was placed close to the casts for calibration.

#### 2.2.4. Visibility Assessment and Measurements

Visibility of the extrahepatic ducts, GB, cystic duct, CBD, and pancreatic duct were descriptively assessed in MRCP, FRCP, and corrosion casting on a binary (yes/no) basis. Diameter measurements in MRCP, FRCP, and corrosion casts were performed using the same protocol, technique, and equipment or devices as those published for cat cadavers in a recent study [13]. Measurements of structures in MRCP, FRCP, and corrosion casts were performed in the same dorsal plane and at the same standardized measure points (Figure 1) using digital image analysis software ImageJ 1.52 (Bethesda, MD, USA). The MRCP measurements were performed by the first author (VR, GB measurements) and by the fourth author (DH, measurements in the cystic duct, CBD, extrahepatic ducts, and pancreatic ducts). Measurements in FRCP were performed by the first author (VR) and in corrosion casting by the third author (SPA).

Diameter measurements were taken from the lumen of the ducts and GB (inner diameter; not including the wall itself) as follows: (a) left and right extrahepatic ducts—at the beginning of the ducts entering the CBD; (b, c, d) GB in three sites—fundus, body, and neck; (e) cystic duct— leaving the GB; (f, g) CBD—at extrahepatic ducts’ junction and at major duodenal papilla; and (h) right and left pancreatic ducts—at the beginning of the ducts leaving the accessory pancreatic duct (Figure 1).

### 2.3. Statistical Analyses

Statistical analyses were performed by a statistician using GraphPad Prism 9.0.0 software. Due to the small sample size of the study, non-parametric tests were applied, and data are presented as median (range). The diameters of the extrahepatic ducts, GB, cystic duct, CBD, and pancreatic ducts in MRCP, FRCP, and corrosion casting were compared by Wilcoxon test. Concordance between MRCP and FRCP and between MRCP and corrosion casting was assessed using Lin’s concordance correlation coefficient with the corresponding 95% confidence interval (CI) [23]. If the null value is not included in the 95% CI, the concordance between the two measurements is considered statistically significant. Bland–Altman plots were generated to depict the agreement between MRCP and FRCP [24]; median differences and 95% limits of agreement (LoA) were calculated. If the null difference is not included in the 95% LoA, the agreement between the two measurements is considered statistically significant. The relationship between MRCP and corrosion casting was assessed using the Spearman correlation coefficient. The following scale was employed for the degree of correlation: r = 0.00–0.19, very weak; r = 0.20–0.39, weak; r = 0.40–0.59, moderate; r = 0.60–0.79, strong; and r = 0.80–1.0, very strong [25]. *p* ≤ 0.05 was considered statistically significant.

## 3. Results

### 3.1. Dogs

All eight dogs successively underwent MRCP, FRCP, corrosion casting, gross pathology, and histopathology. The signalment and histopathological findings are shown in Table 2. According to the histopathological assessments, six dogs (Dogs 1–6) had no evidence of biliary or pancreatic disorders, one dog (Dog 7) had pancreatic pathology, and one dog (Dog 8) had biliary disorders.

### 3.2. Visibility and Measurements

The GB (fundus, body, and neck), cystic duct, and CBD (at papilla and at extrahepatic ducts’ junction) were visible in MRCP, FRCP, and corrosion casting in all eight dogs. Visual comparisons between MRCP, FRCP and corrosion casting in Dog 6 with no evidence of disorders and Dog 8 with biliary disorders are shown in Figure 2 and Figure 3, respectively. The extrahepatic ducts and pancreatic ducts were variably visible in the MRCP and FRCP images. This did not allow for statistical analysis, only descriptive assessment.

The diameter measurements for the right and left extrahepatic ducts, GB, cystic duct, CBD, and right and left pancreatic ducts of the dogs with normal biliary tracts (Dogs 1–7) are summarized in Table 3. In these seven dogs, the MRCP measurement values for the GB, cystic duct, and CBD were consistently lower than those measured by FRCP. The difference was significant for the GB fundus, GB body, cystic duct, and CBD at the extrahepatic ducts’ junction. When the diameter measurements from MRCP and corrosion casting were compared, the MRCP measurement values for the GB fundus, GB body, CBD at papilla, and CBD at the extrahepatic ducts’ junction tended to be lower than those of corrosion casting; however, the difference was not significant.

The diameter of the extrahepatic ducts, GB (fundus, body, and neck), cystic duct, CBD (at papilla and at extrahepatic ducts’ junction), and pancreatic ducts of Dog 8 are presented in Table 4.

### 3.3. Agreement and Concordance between Measurements in MRCP and FRCP

There was a statistically significant concordance between MRCP and FRCP in measuring the GB and cystic duct. The strongest concordance was in measuring the GB neck (0.91; 95% CI 0.62–0.98). The concordance between MRCP and FRCP in measuring the CBD was not statistically significant (Table 5). Bland–Altman plots indicated statistical agreement between MRCP and FRCP for measuring the GB fundus, GB body, CBD at papilla, and CBD at the extrahepatic ducts’ junction as the null value was included in the 95% LoA (Figure 4A,B,E,F). On the other hand, Bland–Altman plots showed that MRCP and FRCP were not statistically comparable for the GB fundus, GB body, and CBD at the extrahepatic ducts’ junction, as all seven measurements were below the null value (Figure 4A,B,F).

### 3.4. Comparison between Measurements in MRCP and Corrosion Casting

Lin’s concordance correlation coefficient showed statistically significant concordance between MRCP and corrosion casting in measuring all biliary structures except for the GB body, with the strongest concordance for the GB neck (0.94; 95% CI 0.71–0.99) (Table 5).

The Spearman correlation coefficient indicated very strong correlations between MRCP and corrosion casting for measuring the GB neck and CBD at papilla (r = 0.98 and 0.86, respectively; *p <* 0.05). Correlations between MRCP and corrosion casting for measuring the GB fundus (r = 0.74, *p* < 0.05), GB body (r = 0.62), cystic duct (r = 0.76, *p* < 0.05), and CBD (r = 0.60) at the extrahepatic ducts’ junction were strong.

## 4. Discussion

Despite the existence of studies examining MR images and MRCP of the biliary tract and pancreatic ducts in healthy and diseased cats [11,12,26], there are no published investigations on MRCP in dogs. We compared the appearance and measurements of the biliary tract and pancreatic ducts acquired by MRCP with those obtained by post-mortem FRCP and corrosion casting. Our study confirmed that MRCP is feasible in dogs and can be considered as a reliable modality for visualizing and measuring the diameter of the extrahepatic biliary tract for structures >0.95 mm. In human medicine, direct biliary and pancreatic duct imaging in patients has historically been dependent on ERCP with direct catheter-based infusion of intraductal contrast, and ERCP is still considered the reference standard for ductal imaging [27]. For veterinary medicine, our study showed that MRCP can visualize the same morphological features of the biliary tree as shown by post-mortem FRCP and in corrosion casting in dogs. Previous studies have shown that ERCP is technically demanding in live dogs, with a success rate between 66% and 87%. The risk of complications seems to be low but should be considered when planning ERCP in canine patients [16,17]. MRCP is a less invasive alternative to diagnostic ERCP and may be a beneficial imaging tool to distinguish patients that require therapeutic ERCP from those requiring abdominal surgery or medical intervention.

Diameters measured in MRCP were significantly smaller than those of FRCP in most of the measurement points, while diameters measured in MRCP were non-significantly smaller than those of corrosion casting. As a radiographic method, FRCP is subject to magnification, which can be addressed by using radiopaque markers of defined size to estimate the degree of magnification. A study in humans showed that ERCP overestimates the size of the pancreatic duct by an average of 1.5 times with respect to MRCP [28], as the injected contrast material expands the biliary system. This results in larger diameters than those measured in MRCP. This phenomenon also occurs in corrosion casting when the injected VPS enlarges the biliary tract and the pancreatic ducts. However, our results revealed that the injection of contrast during FRCP is more intense than the expanding impact of VPS injected into the biliary tract and pancreatic ducts during corrosion casting. These findings are consistent with a previous post-mortem study on 12 cats, where diameter measurements in MRCP were significantly smaller than those in FRCP and corrosion casting [13].

In the current study, the extrahepatic ducts were visible to various degrees. However, pancreatic ducts could be identified in the MRCP images of only one dog with a body weight of 18 kg. An ultrasonographic study on 242 adult dogs (body weight 1.4–55 kg) showed that the diameters of pancreatic ducts ranged between 0.4 and 1.2 mm, and the diameter significantly increased with body weight [29]. In a recent post-mortem MRCP study on cats, the minimum measurable duct diameter was 0.95 mm [13]. Based on these studies, we assumed that the pancreatic ducts of adult dogs would be visualized better than the ones in cats with a body weight ≤5 kg. The method used in the present study may not have been sensitive enough for pancreatic structures. MRI devices with a magnet strength of 3.0 Tesla produce more sophisticated images [30]. In addition, the low body temperature in post-mortem studies affects signal intensity, leading to a lower image contrast [31,32]. Moreover, the lack of biliary and pancreatic juice flow post-mortem likely has a negative effect on the visibility of the biliary tract and pancreatic ducts in MRCP. Prior studies in cats with and without cholangitis and pancreatitis [11,12] and in a study in children [33] showed that secretin enhances MRCP visualization of the pancreatic ducts due to its dilatory effect. These findings suggest the need for secretin while performing MRCP in live animals to optimize the detection of fine biliary and pancreatic ducts. Another aspect that should be considered in MRCP of live animals is the control of motion artifacts since imaging quality is a significant factor regarding the ability of MRCP to reliably detect ductal anatomy [27].

The maximum diameters of the CBD at papilla and extrahepatic ducts’ junction measured by MRCP in the seven dogs without disorders of the biliary tract were 2.27 and 3.50 mm, respectively. Measurement of the CBD has been used to evaluate obstruction of the biliary system. The mean diameter of a normal CBD at its distal part as measured by ultrasonography has been proposed to be <3 mm [34]. However, that study was performed in only three dogs weighing 14 to 20 kg. Therefore, a 3 mm cut-off value may not be a clear and reliable criterion across various dog breeds and body weights. The CBD diameter in seven healthy beagle dogs with a mean body weight of 14.1 ± 3 kg was earlier measured in ERCP with a measurement of 3.04 ± 1.89 mm at the duodenal papilla [16]. Recently, the CBD diameter at the porta hepatis, duodenal papilla level, and mid-portion was assessed using computed tomography in 283 healthy dogs weighing 1.5 to 29.5 kg [35]. The age of the dogs in that study varied between 0.5 and 17 years (mean 8.8 years). No correlation was found between CBD diameter and dog age, but there was a significant difference in the CBD diameter at each measurement point between body weight groups [35]. Furthermore, the body weight and CBD diameter showed a positive linear correlation at each point. Thus, the normal reference ranges of CBD diameter should be applied for different body weights. The discrepancy between the diameter measurements of the CBD obtained in our study and the previously reported measurements is likely due to the fact that the measurements were performed by different imaging modalities in dogs with varying body weights. Even though age of the dog has not been found to be related to the CBD diameter [35], the effect of old age on the measurements remains unknown. The youngest dog in our study was 9 years old. Dog 8 with cholangitis and cholestasis exhibited a dilated GB neck in MRCP, FRCP, and corrosion casting when compared with the seven dogs without disorders of the biliary tract. In a study of 45 dogs with cholangitis confirmed by histopathology, ultrasonography showed a distended GB in 24 dogs (53.3%) [1]. In a retrospective study, ultrasonography revealed distended GB in three of seven dogs with various types of cholangitis; one out of those three dogs had concurrent biliary stasis [17].

Our study had several limitations, primarily because it was performed post-mortem which resulted in a lower chance of displaying the biliary tree and pancreatic ducts in MRCP due to the lack of biliary and pancreatic juice flow. The low body temperature can lead to a lower image contrast in post-mortem MRI due to its influence on signal intensity [30]. In addition, the study was performed using a 1.5 T machine, which is not optimal for the visualization of tiny structures. All dogs in this study were presented only for euthanasia, and therefore, no detailed history of the animals’ health status was available, and the body condition score was not recorded in the clinical files or autopsy reports. Future studies need to take the difference between body weight and body condition scores into consideration when comparing hepatobiliary and pancreatic duct diameters of dogs of different sizes. In the current study, gross pathologic evaluation, histopathology, and the use of corrosion casting provided information to confirm and validate the MRCP findings. 

Typical to proof-of-concept studies [36], the small number of dogs caused difficulties in applying statistical tests, but an anatomical comparison of the images provided descriptive data that further supported the quantitative results. Additionally, the dogs in the present study provided good material for evidence of the feasibility of MRCP, as not only normal dogs but also dogs with pancreatobiliary disorders were included.

## 5. Conclusions

This observational, proof-of-concept study on the bodies of eight dogs revealed that MRCP is feasible in dogs and allows for the visualization of and measurements from the extrahepatic biliary tree. However, ductal structures of <1 mm, such as pancreatic ducts, were difficult to visualize using a 1.5 Tesla MRI magnet. Future studies should prospectively validate the feasibility of MRCP in live dogs and provide reference diameters of the biliary tract and pancreatic ducts in dogs with different body weights and ages and various biliary and pancreatic disorders.

## Figures and Tables

**Figure 1 animals-13-02517-f001:**
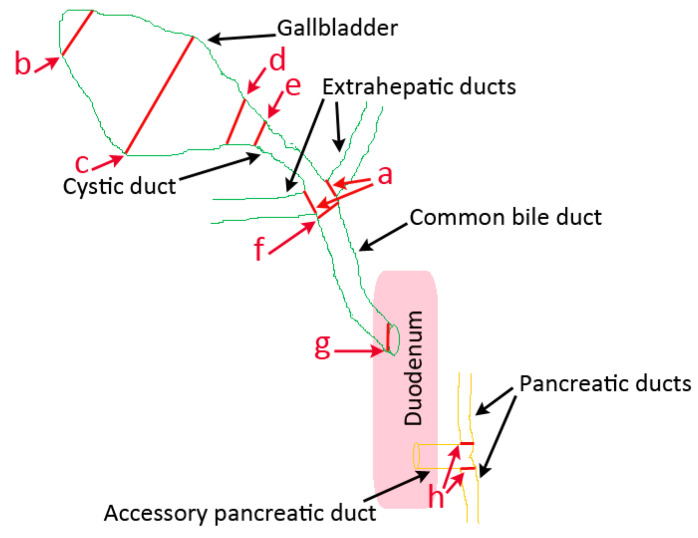
Schematic representation of the canine pancreatobiliary tract illustrating the sites where diameter measurements were made in MRCP, FRCP, and corrosion casting (red lines). a: right and left extrahepatic ducts; b: gallbladder fundus; c: gallbladder body; d: gallbladder neck; e: cystic duct; f: common bile duct at extrahepatic ducts’ junction; g: common bile duct at duodenal papilla; h: right and left pancreatic ducts.

**Figure 2 animals-13-02517-f002:**
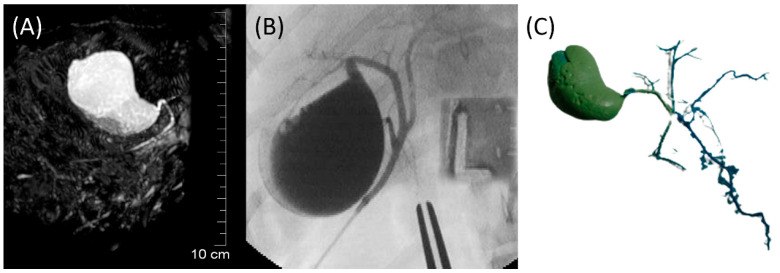
Images of Dog 6 with no evidence of biliary and pancreatic disorders. (**A**) 3D-TSE-MRCP image showing normal gallbladder, cystic duct, and common bile duct. (**B**) FRCP image demonstrating normal biliary tract filled with contrast material. (**C**) Corrosion cast demonstrating normal biliary tract.

**Figure 3 animals-13-02517-f003:**
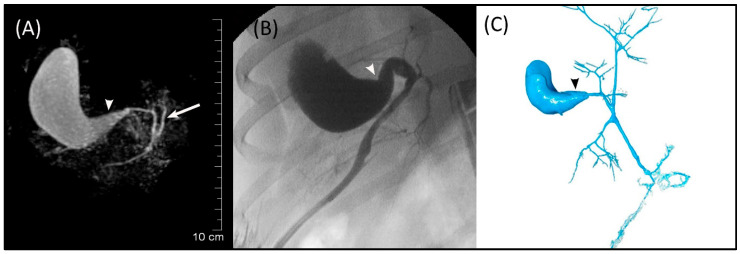
Images of Dog 8 with acute extrahepatic cholestasis and focal destructive cholangitis. (**A**) 3D-TSE-MRCP illustrating dilated gallbladder neck (arrowhead). Normal left extrahepatic duct is visible (arrow). (**B**) FRCP image showing dilated gallbladder neck (arrowhead). (**C**) Corrosion cast demonstrating dilated gallbladder neck (arrowhead).

**Figure 4 animals-13-02517-f004:**
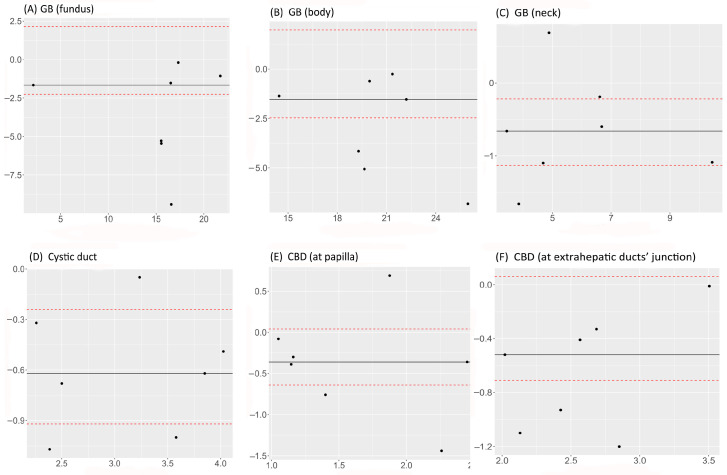
Bland–Altman plots comparing MRCP with FRCP in measuring diameters of the GB fundus (**A**), GB body (**B**), GB neck (**C**), cystic duct (**D**), CBD at papilla (**E**), and CBD at extrahepatic ducts’ junction (**F**). The *x*-axes show the median of both measurements, the *y*-axes show the difference between the two measurements, the dashed lines represent the 95% CI, and the continuous lines represent the median difference.

**Table 1 animals-13-02517-t001:** List of sequences and corresponding parameters applied for MRI scanning of the 8 dogs.

Sequence Number	Sequence Type	Sequence Parameters TR/TE (ms) Slice Thickness (mm) Acquisition Voxel Size (mm^3^)
1	Transverse T1-w-TSE	400–700/10
		4
		1 × 1 × 4
2	Dorsal T1-w-TSE	400–700/10
		4
		1 × 1 × 4
3	Transverse T2-w-TSE	3000/100
		4
		1 × 1 × 4
4	Dorsal T2-w-TSE	3000/100
		4
		1 × 1 × 4
5	Transverse T2-w-TSE-SPIR	3000/100
		4
		1 × 1 × 4
6	Dorsal T2-w-TSE-SPIR	3000/100
		4
		1 × 1 × 4
7	Dorsal 3D-TSE-MRCP	2500/600
		-
		0.9 × 1 × 1.1

T1-w-TSE: T1-weighted turbo spin echo; T2-w-TSE: T2-weighted turbo spin echo; TR: repetition time; TE: echo time; SPIR: with fat saturation; 3D-TSE-MRCP: three-dimensional turbo spin echo magnetic resonance cholangiopancreatography.

**Table 2 animals-13-02517-t002:** Breed, age, sex, weight, and histopathological findings of 8 dogs.

Dogs	Breed	Age (Years)	Sex	Weight (kg)	Histopathological Findings
1	Mixed	15	F	7.1	NA
2	Mixed	13	(M)	10.8	NA
3	Staffordshire Bull Terrier	9	(F)	18.1	NA
4	Mixed	14	F	23.0	NA
5	Spanish Water Dog	14	M	15.1	NA
6	Mixed	11	F	10.3	NA
7	Mixed	10	(F)	15.4	Moderate pancreatic fibrosis
8	French Bulldog	14	F	17.1	Acute extrahepatic cholestasis, Focal destructive cholangitis
	Median (range)	13.5 (9–15)		15.3 (7.1–23)	

F = female, (F) = spayed female, M = male, (M) = neutered male, NA = no abnormalities.

**Table 3 animals-13-02517-t003:** Diameter measurements of the gallbladder (GB), cystic duct, common bile duct (CBD), extrahepatic ducts, and pancreatic ducts as determined by magnetic resonance cholangiopancreatography (MRCP), fluoroscopic retrograde cholangiopancreatography (FRCP), and corrosion casting in dogs without disorders of the biliary tract (Dogs 1–7).

Structure	Measurement			Statistical Significance of the Difference between Measurements	
	MRCP (mm)	FRCP (mm)	Corrosion Casting (mm)	*p*-Value (MRCP vs. FRCP)	*p*-Value (MRCP vs. Corrosion Casting)
GB (fundus)	12.90 (1.32–21.18)	18.18 (2.98–22.25)	17.25 (2.01–23.80)	0.016 *	0.110
GB (body)	19.66 (13.77–22.54)	21.49 (15.13–29.36)	21.35 (14.05–29.26)	0.016 *	0.160
GB (neck)	5.23 (3.03–9.89)	5.24 (3.78–10.98)	4.45 (2.15–10.00)	0.110	0.690
Cystic duct	3.08 (1.85–3.78)	3.26 (2.42–4.27)	2.73 (2.00–4.71)	0.016 *	0.220
CBD (at papilla)	1.02 (0.95–2.27)	1.53 (1.09–2.98)	1.52 (0.88–2.88)	0.160	0.670
CBD (at extrahepatic ducts’ junction)	2.25 (1.58–3.50)	2.85 (2.28–3.51)	2.68 (1.32–4.58)	0.016 *	0.110
Extrahepatic duct (right)	Visible in 4/7 dogs 1.25 (1.12–1.46)	Visible in 6/7 dogs 1.40 (1.23–2.89)	1.30 (1.02–2.21)	NP	NP
Extrahepatic duct (left)	Visible in 4/7 dogs 1.27 (0.96–1.46)	Visible in 7/7 dogs 1.56 (1.53–2.98)	1.51 (1.24–2.74)	NP	NP
Pancreatic duct (right)	Visible in 1/7 dogs 1.12	Visible in 4/7 dogs 1.43 (0.74–1.52)	1.02 (0.26–1.45)	NP	NP
Pancreatic duct (left)	Visible in 1/7 dogs 0.98	Visible in 4/7 dogs 1.21 (1.01–1.80)	1.49 (0.08–2.20)	NP	NP

Data are presented as median (range). *p* < 0.05 is considered significant. *: statistically significant difference. NP: not performed.

**Table 4 animals-13-02517-t004:** Diameter measurements of the gallbladder (GB), cystic duct, common bile duct (CBD), extrahepatic ducts, and pancreatic ducts as determined by magnetic resonance cholangiopancreatography (MRCP), fluoroscopic retrograde cholangiopancreatography (FRCP), and corrosion casting in Dog 8 with disorders of the biliary tract.

Structure	Measurement		
	MRCP (mm)	FRCP (mm)	Corrosion Casting (mm)
GB (fundus)	10.87	12.24	9.00
GB (body)	19.48	21.32	20.00
GB (neck)	15.69	19.21	18.13
Cystic duct	1.94	3.32	2.21
CBD (at papilla)	1.82	2.87	2.00
CBD (at extrahepatic ducts’ junction)	2.10	3.45	3.11
Extrahepatic duct (right)	1.16	1.22	1.00
Extrahepatic duct (left)	1.45	2.65	2.67
Pancreatic duct (right)	NV	NV	0.90
Pancreatic duct (left)	NV	NV	1.00

NV: not visible.

**Table 5 animals-13-02517-t005:** Lin’s concordance correlation coefficient (and 95% CI) between MRCP and FRCP and between MRCP and corrosion casting in measuring biliary tract diameters in dogs without biliary tract disorders (Dogs 1–7).

Structure	Concordance between MRCP and FRCP (95% CI)	Concordance between MRCP and Corrosion Casting (95% CI)
GB (fundus)	0.73 * (0.20–0.93)	0.76 * (0.23–0.95)
GB (body)	0.57 * (0.04–0.85)	0.39 (–0.23–0.79)
GB (neck)	0.91 * (0.62–0.98)	0.94 * (0.71–0.99)
Cystic duct	0.65 * (0.16–0.88)	0.78 * (0.31–0.94)
CBD (at papilla)	0.43 (–0.26–0.83)	0.74 * (0.13–0.95)
CBD (at extrahepatic ducts’ junction)	0.37 (–0.08–0.69)	0.56 * (0.01–0.85)

* Statistically significant concordance.

## Data Availability

The data that support the findings of this study are available from the corresponding author, upon reasonable request.

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
