# Peer review of "Feasibility of Magnetic Resonance Cholangiopancreatography in Dogs—A Post-Mortem Study"

_animals, 2023, doi:10.3390/ani13152517_

Round 1

Reviewer 1 Report

The authors compared the results of cholangiopancreatic MR imaging with retrograde fluoroscopy and corrosion casting in 8 geriatric canine cadavers (6 without hepatobiliary or pancreatic disease) diameter measurements were correlated with all three and MR cholangiopancreatography was found to visualize ducts > 1mm in size.

Overall the manuscript was well organized and well written. I have just a few notes and suggestions to strengthen the quality of the manuscript.

Pancreatic ducts are less than 1 mm which is too small to visualize using 1.5 tesla magnet for MR imaging. The authors previously published that information. It is not clear why the authors included information about the pancreas in this study (in the introduction and methods) when they did not have a system sensitive enough to measure these ducts in 6/7 animals. This is an error in the study design that needs to be resolved.

Consequently, they were not able to compare and/or validate the correlation between MR and the other modalities. The authors cannot draw conclusions on MR pancreatography given the lack of data. The paper should be revised to omit the pancreas and indicate a study of cholangiography/biliary imaging with the 1.5 tesla magnet, repeat the study with a different MR system OR change the focus of the paper and discussions to correctly reflect that only biliary structures correlated between MR and the other imaging methods.

The simple summary does not meet the journal criteria, as it includes both jargon and acronyms, this should be revised to meet standards.

Introduction.

The 1st sentence in the introduction should be rewritten or include a reference.

The 1st “and” should be deleted and replaced with a comma in sentences like in line 49 and line 73.

The second paragraph should be omitted. It is not relevant to the study design (which doesn’t include ultrasonography) if the authors wish to include this information, they should add ultrasound measurements to the study design.

Methods.

 It is unclear if any of these animals were normal healthy adults. The health status and presence of any co-morbidities should be included in the demographic information. The absence of any overt biliary disease, does not indicate that canine had a normal biliary system.

The authors included only senior/geriatric dogs in the study. Either younger adults should be included to provide a more representative sample population or the paper should be revised to indicate these results are for senior dogs only.

The section on FRCP needs to be expanded to include additional details. Enough detail should be included to allow for reasonable repeatability. For example: how was the amount of contrast used determined? How mush pressure was applied? How were these standardized? How did you know when ducts were filled, but the parenchyma was not?

Who made all of the measurements? Were they blinded as to the animal? Repeated measurements should have been made to determine the repeatability and error between subsequent measurements. If these were conducted by different individuals (as the contribution section indicated) then the authors must account for inter-observer variability?

The authors should add comparisons/agreement between FRCP and corrosion casting and the associated p-values for each structure.

Results

Not enough information is provided in table 2. Include what breeds were represented in the mixed breeds and what was each animal’s body condition score? It is difficult to interpret the relevance of the weight information for especially the mixed breeds (as written) because we don’t know if they are large or small for their breed.

Add comparisons/agreement between FRCP and corrosion casting and the associated p-values for each structure in Table 3.

Add concordance between FRCP and corrosion casting and the associated p-values for each structure in Table 5.

Figure 4 and table 6 feel redundant and an less exact ways of showing information already listed in other figures/tables. Both of these figures/tables should be deleted or relevance/value explained.

Discussion

Line 367 to 374 should be reworded or revised for clarity.

Line editing/numbers did not format correctly with the information in figure 6 (lines 301-305)

References.

Journal article titles in reference 1 and 5 are capitalized which is not consistent with the rest of the references.

Consider alternatives to minimize the use of self-citations.

Few minor grammar and syntax errors (example: repeated use of AND in the same sentence).

Author Response

Response to Reviewer 1 Comments

The authors compared the results of cholangiopancreatic MR imaging with retrograde fluoroscopy and corrosion casting in 8 geriatric canine cadavers (6 without hepatobiliary or pancreatic disease) diameter measurements were correlated with all three and MR cholangiopancreatography was found to visualize ducts > 1mm in size.

Overall the manuscript was well organized and well written. I have just a few notes and suggestions to strengthen the quality of the manuscript.

Point 1: Pancreatic ducts are less than 1 mm which is too small to visualize using 1.5 tesla magnet for MR imaging. The authors previously published that information. It is not clear why the authors included information about the pancreas in this study (in the introduction and methods) when they did not have a system sensitive enough to measure these ducts in 6/7 animals. This is an error in the study design that needs to be resolved. Consequently, they were not able to compare and/or validate the correlation between MR and the other modalities. The authors cannot draw conclusions on MR pancreatography given the lack of data. The paper should be revised to omit the pancreas and indicate a study of cholangiography/biliary imaging with the 1.5 tesla magnet, repeat the study with a different MR system OR change the focus of the paper and discussions to correctly reflect that only biliary structures correlated between MR and the other imaging methods.

Response 1: Our previously published study included 12 cats with body weights between 2.7 and 5.1 kg. In a study by Penninck et al. (https://pubmed.ncbi.nlm.nih.gov/23438119/) which comprised 242 adult dogs with body weights ranging from 1.4 to 55 kg, it has been shown that diameter of the pancreatic ducts, assessed ultrasonographically, significantly increased with body weight. Based on that study, we assumed that in our current investigation performed on dogs, pancreatic ducts would be visualized better than compared the ones in cats as the body weights of the dogs’ population were higher than cats (7.1-23 kg in our dogs vs 2.7-5.1 kg in the previous study on cats). Therefore, it was worth to perform MR cholangiopancreatography in dogs to see if the pancreatic ducts are visible.

Point 2: The simple summary does not meet the journal criteria, as it includes both jargon and acronyms, this should be revised to meet standards.

Response 2: Thank you. Revised as suggested.

Introduction

Point 3: The 1st sentence in the introduction should be rewritten or include a reference.

Response 3: Thank you for the comment. Two relevant references were included. (Introduction, line 45; References 1 and 2)

Point 4: The 1st “and” should be deleted and replaced with a comma in sentences like in line 49 and line 73.

Response 4: Partially revised as suggested. However, deleting the first “and” and replacing it with a comma would change the meaning of the sentence. (Introduction, line 50)

Point 5: The second paragraph should be omitted. It is not relevant to the study design (which doesn’t include ultrasonography) if the authors wish to include this information, they should add ultrasound measurements to the study design.

Response 5: The second paragraph has been omitted as suggested.

Methods

Point 6: It is unclear if any of these animals were normal healthy adults. The health status and presence of any co-morbidities should be included in the demographic information. The absence of any overt biliary disease, does not indicate that canine had a normal biliary system.

Response 6: All eight dogs included in this study came to the hospital only for euthanasia. There is no information or history about the health status of the dogs. However, we explained in the manuscript that cadavers of the dogs were examined in autopsy, and based on gross and histopathological examination, which is a strong support, the health condition of the biliary system and pancreas were reported.

Point 7: The authors included only senior/geriatric dogs in the study. Either younger adults should be included to provide a more representative sample population or the paper should be revised to indicate these results are for senior dogs only.

Response 7: The manuscript has been revised to indicate the old age of the dogs as suggested. (Discussion, lines 367-369; Conclusion, line 394)

Point 8: The section on FRCP needs to be expanded to include additional details. Enough detail should be included to allow for reasonable repeatability. For example: how was the amount of contrast used determined? How mush pressure was applied? How were these standardized? How did you know when ducts were filled, but the parenchyma was not?

Response 8: We thank the Reviewer for the comment.

Unfortunately, it was not possible to determine the exact amount of contrast medium needed for filling the biliary tract and pancreatic duct, as part of contrast medium was leaking out from papilla while it was injected into the ducts. However, according to previous publications determining the volume of the gallbladder in dogs with different weights (https://pubmed.ncbi.nlm.nih.gov/17259453/)(https://www.ncbi.nlm.nih.gov/pmc/articles/PMC4591159/), we were able to estimate how much contrast medium was needed for each dog.

The contrast medium was injected with manually controlled pressure using a previously published approach for performing ERCP in dogs (https://pubmed.ncbi.nlm.nih.gov/15869151/). In order to avoid parenchyma filling, the injection was stopped when contrast medium filled the gallbladder, or the proximal part of the pancreatic ducts were clearly visible. The manuscript has been revised for more clarification as suggested (Materials and methods, lines 134-135).

Point 9: Who made all of the measurements? Were they blinded as to the animal? Repeated measurements should have been made to determine the repeatability and error between subsequent measurements. If these were conducted by different individuals (as the contribution section indicated) then the authors must account for inter-observer variability?

Response 9: Measurements in each method were performed by an expert in the field. Measurements in the FRCP images were performed by the first author (VR). Measurements in the MRCP images were performed by VR (GB measurements) and DH (measurements of the cystic duct, CBD, extrahepatic ducts, and pancreatic ducts), and the measurements regarding corrosion casting were performed by fourth author (SPA). DH was blinded, while VR and SPA were not blinded. Our setting did not allow assessment of inter-observer variability. To avoid confusion between the text and the contribution section, this aspect has been clarified (Materials and methods, lines 171-175).

Point 10: The authors should add comparisons/agreement between FRCP and corrosion casting and the associated p-values for each structure.

Response 10: We thank the Reviewer for this comment. The same aspect was already thoroughly considered by the authors before submitting the manuscript. We respectfully disagree with the suggestion to add comparison/agreement between FRCP and corrosion casting. Comparison/agreement between FRCP and corrosion casting is beyond the scope of our study and would not add significant value to it. Additionally, corrosion casting is not a diagnostic technique, and its main role in the current study is to confirm the morphology of the biliary tract/pancreatic ducts. For example, corrosion casting allowed us to conclude that although in some MRCP and FRCP studies the hepatic ducts or the pancreatic ducts were not visible, they were still present.

Results

Point 11: Not enough information is provided in table 2. Include what breeds were represented in the mixed breeds and what was each animal’s body condition score? It is difficult to interpret the relevance of the weight information for especially the mixed breeds (as written) because we don’t know if they are large or small for their breed.

Response 11: We respectfully disagree with this suggestion. The objective of our study was to assess the overall feasibility of MRCP in dogs, and the weight range of the dogs sufficiently shows their size for this purpose. Based on the information in the patient recording system, it is not possible to have further details of breeds represented in mixed breeds or of body condition score of the animals.

Point 12: Add comparisons/agreement between FRCP and corrosion casting and the associated p-values for each structure in Table 3.

Response 12: Please see the response to Point 10.

Point 13: Add concordance between FRCP and corrosion casting and the associated p-values for each structure in Table 5.

Response 13: Please see the response to Point 10.

Point 14: Figure 4 and table 6 feel redundant and an less exact ways of showing information already listed in other figures/tables. Both of these figures/tables should be deleted or relevance/value explained.

Response 14: We thank the Reviewer for the comment. The manuscript was revised to include the important information of the Table 6 in the text, and Table 6 was deleted (Results, lines 293-297). However, we consider Figure 4 informative and want to keep it in the manuscript.

Discussion

Point 15: Line 367 to 374 should be reworded or revised for clarity.

Response 15: Revised as suggested (Discussion, lines 356-364).

Point 16: Line editing/numbers did not format correctly with the information in figure 6 (lines 301-305)

Response 16: Revised.

Point 17: Journal article titles in reference 1 and 5 are capitalized which is not consistent with the rest of the references.

Response 17: Revised.

Point 18: Consider alternatives to minimize the use of self-citations.

Response 18: There are very few publications applying ERCP and MRCP techniques in dogs and cats. All of them are referred in this manuscript. One self-citation was removed (https://www.ncbi.nlm.nih.gov/pmc/articles/PMC4591159/) along with deleting the second paragraph of the Introduction. Two additional references on the use of MRCP in humans were added (References number 8 and 9).

Comments on the Quality of English Language

Point 19: Few minor grammar and syntax errors (example: repeated use of AND in the same sentence).

Response 19: Thank you for this comment. We have tried to correct the errors.

Reviewer 2 Report

In this original article, the feasibility of Magnetic Resonance Cholangiopancreatography (MRCP) in dogs was investigated. The study aimed to compare the visibility and measured diameters of the biliary tract and pancreatic ducts using MRCP with fluoroscopic retrograde cholangiopancreatography (FRCP) and corrosion casting. Eight adult dogs were included, with six dogs having no hepatobiliary disorders, one with pancreatic pathology, and one with biliary pathology. It should be pointed out that this study is very interesting and provides a high level of novovelty. Material and methods section along with provided table and figures are very informative and carefully prepared. The introduction part is well-presented and fully justifies the topic of undertaken research study. The design of the study is precisely and well conducted, the sample size of each experimental group is adequate. The obtained research results undoubtedly add to existing knowledge. Basically all used literature items are properly selected to the topic of the work and are up-to-date. Overall I had a great pleasure reading this manuscript, as it presents a novel point of view and is very carefully prepared.

Author Response

We thank the Reviewer for positive comments on our study.

Reviewer 3 Report

The manuscript "Feasibility of magnetic resonance chplangiopancreatography in dogs - a post-mortem study" submitted for review is an important contribution to the discussion on the diagnosis of bile ducts and pancreatic ducts in dogs. The research material consisted of 8 cadavers, including 6 healthy and one with pancreatic and biliary disorders. This is a significant limitation of the study, significantly limiting the credibility of the results. Suggested by the authors, future intravital studies should be performed on more numerous samples, taking into account racial and pathological factors. The concept of the work, the selection of research methods, including statistical methods, do not raise any doubts. The discussion is based on properly selected references and leads to clearly formulated conclusions. I believe that the work deserves publication, after revision with a few remarks:

The number of animals used for each method varies, as can be seen from the reported results. In section 2.1, the authors declare that all 8 dogs were included in the study, so why are the results presented in table 5 only applicable to 7? Please refer to the negative values of concordance correlation coefficients presented in table 5 in the discussion. What could be the reason? The authors willingly use abbreviations concerning diagnostic methods and anatomical structures. Unfortunately, this does not contribute to the clarity of the article and is the cause of some inaccuracies. Endoscopic retrograde cholangiopancreatography (ERCP) is described in detail in the Introduction chapter, followed by the declaration of FRCP in Materials and Methods and later. In the discussion, these abbreviations are also used interchangeably. Please explain and sort it out. Anatomic abbreviations are applied to some structures and not to others. Please use full names consistently. Anatomical directional terms like "proximal", "distal" seem to be more readable than "beginning" or "at junction". Use "inner diameter" instead of "from lumen of the ducts".

Author Response

Response to Reviewer 3 comments

The manuscript "Feasibility of magnetic resonance chplangiopancreatography in dogs - a post-mortem study" submitted for review is an important contribution to the discussion on the diagnosis of bile ducts and pancreatic ducts in dogs. The research material consisted of 8 cadavers, including 6 healthy and one with pancreatic and biliary disorders. This is a significant limitation of the study, significantly limiting the credibility of the results. Suggested by the authors, future intravital studies should be performed on more numerous samples, taking into account racial and pathological factors. The concept of the work, the selection of research methods, including statistical methods, do not raise any doubts. The discussion is based on properly selected references and leads to clearly formulated conclusions. I believe that the work deserves publication, after revision with a few remarks:

Point 1: The number of animals used for each method varies, as can be seen from the reported results. In section 2.1, the authors declare that all 8 dogs were included in the study, so why are the results presented in table 5 only applicable to 7?

Response 1: We thank the Reviewer for the question. In table 5, we wanted to show the concordance between MRCP and FRCP, as well as, between MRCP and corrosion casting. As one of the dogs showed evidence of biliary disorders in histopathology examination, the 7 dogs without biliary tract disorders were considered for comparison to avoid the impact of possibly increased biliary tract diameter on concordance correlation between modalities. This decision was also discussed with the statistician.

Point 2: Please refer to the negative values of concordance correlation coefficients presented in table 5 in the discussion. What could be the reason?

Response 2: Based on our statistician, the Lin's CCC can assume values in the interval [-1 to 1]. For each case, together with the pointwise estimation, the 95% confidence interval is also presented. If 0 is included in the 95% confidence interval, then we cannot reject for a fixed level of alpha=0.05 the null hypothesis stating that there is no concordance between the two considered variables. With the available data, this applies to 3 cases in Table 5. In general, the 95% confidence intervals of Table 5 are broad since the sample size is small (n=7). Therefore, the lower limit of the confidence interval might be negative. The confidence interval will shrink as the sample size increases, i.e. leading to smaller uncertainties around the pointwise estimate.

Point 3: The authors willingly use abbreviations concerning diagnostic methods and anatomical structures. Unfortunately, this does not contribute to the clarity of the article and is the cause of some inaccuracies. Endoscopic retrograde cholangiopancreatography (ERCP) is described in detail in the Introduction chapter, followed by the declaration of FRCP in Materials and Methods and later. In the discussion, these abbreviations are also used interchangeably. Please explain and sort it out.

Response 3: Thank you for the comment. The abbreviations used in this manuscript are established and frequently used in the veterinary medicine’s publications. We prefer to keep the abbreviations to retain the fluency of the paper. In the Discussion, ERCP is used for studies on live objects and FRCP in cadavers. We added post-mortem before FRCP and canine patient after ERCP for avoiding confusions (Discussion, lines 303, 307, 310 and 313).

Point 4: Anatomic abbreviations are applied to some structures and not to others. Please use full names consistently. Anatomical directional terms like "proximal", "distal" seem to be more readable than "beginning" or "at junction". Use "inner diameter" instead of "from lumen of the ducts".

Response 4: Thank you for the comment. According to our response to Point 3, we prefer to keep the abbreviations as they are.

We prefer to keep “at papilla” and “at extrahepatic ducts’ junction” as they reflect the exact sites where measurements were performed.

 “Inner diameter” is now used instead of “from lumen of the ducts” as suggested (Materials and methods, lines 183-184)

Round 2

Reviewer 1 Report

See attached (these can be easily addressed)

Author Response

Response to Reviewer 1 Comments

Point 1: Pancreatic ducts are less than 1 mm which is too small to visualize using 1.5 tesla magnet for MR imaging. The authors previously published that information. It is not clear why the authors included information about the pancreas in this study (in the introduction and methods) when they did not have a system sensitive enough to measure these ducts in 6/7 animals. This is an error in the study design that needs to be resolved. Consequently, they were not able to compare and/or validate the correlation between MR and the other modalities. The authors cannot draw conclusions on MR pancreatography given the lack of data. The paper should be revised to omit the pancreas and indicate a study of cholangiography/biliary imaging with the 1.5 tesla magnet, repeat the study with a different MR system OR change the focus of the paper and discussions to correctly reflect that only biliary structures correlated between MR and the other imaging methods.

Response 1: Our previously published study included 12 cats with body weights between 2.7 and 5.1 kg. In a study by Penninck et al. (https://pubmed.ncbi.nlm.nih.gov/23438119/) which comprised 242 adult dogs with body weights ranging from 1.4 to 55 kg, it has been shown that diameter of the pancreatic ducts, assessed ultrasonographically, significantly increased with body weight. Based on that study, we assumed that in our current investigation performed on dogs, pancreatic ducts would be visualized better than compared the ones in cats as the body weights of the dogs’ population were higher than cats (7.1-23 kg in our dogs vs 2.7-5.1 kg in the previous study on cats). Therefore, it was worth to perform MR cholangiopancreatography in dogs to see if the pancreatic ducts are visible.

Reviewer Counter: It is reasonable for the authors to attempt MR cholangiopancreatography but as I indicated in my review you cannot draw conclusions on MR pancreatography given the lack of data using the MR employed. The paper should be revised and indicate a study of cholangiography/biliary imaging with the 1.5 tesla magnet (you can mention that the method used was not sensitive enough for pancreatic structures), repeat the study with a different MR system (to meet the study objective) OR change the focus of the paper and discussions to correctly reflect that only biliary structures correlated between MR and the other imaging methods. Otherwise in the current form you did not have the appropriate methodology to test the objective as presented in this paper, you can say that you attempted to look at the pancreas but that is all.

Response: We thank the Reviewer for the comment. Use of 1.5 T in our study has been added to the limitations of the study. We respectfully disagree with the opinion of the Reviewer since the offered suggestions for changes do not agree with the objectives of the study. The fact is that MRCP technique was performed in this investigation and not only MRC. However, in the manuscript it was presented that the pancreatic duct was variably visible and concluded that MRCP using 1.5 Tesla magnet on dogs weighing 7.1-23 kg has limitations on visualizing small ducts including extrahepatic and pancreatic ducts. To substantiate our opinion that we provide valuable information to the readers, we extended the discussion about pancreatic ducts elaborating on the unexpected result that pancreatic ducts were not visible in dogs weighing more than 5 kg which is in contrast to the paper of Pennick et al (https://pubmed.ncbi.nlm.nih.gov/23438119/) (Discussion, lines 331-339, Reference no.29).

Point 6: It is unclear if any of these animals were normal healthy adults. The health status and presence of any co-morbidities should be included in the demographic information. The absence of any overt biliary disease, does not indicate that canine had a normal biliary system.

Response 6: All eight dogs included in this study came to the hospital only for euthanasia. There is no information or history about the health status of the dogs. However, we explained in the manuscript that cadavers of the dogs were examined in autopsy, and based on gross and histopathological examination, which is a strong support, the health condition of the biliary system and pancreas were reported.

Reviewer Counter: Euthanasias are more often performed on animals with health issues than normal healthy animals. A statement should be added to the limitations in the discussion that the health The health status and presence of any co-morbidities was unknown. A similar statement should be added to indicate that body condition scoring information was not recorded.

Response: Thank you for the comment. Added as suggested (Discussion, lines 382-388).

Point 10: The authors should add comparisons/agreement between FRCP and corrosion casting and the associated p-values for each structure.

Response 10: We thank the Reviewer for this comment. The same aspect was already thoroughly considered by the authors before submitting the manuscript. We respectfully disagree with the suggestion to add comparison/agreement between FRCP and corrosion casting. Comparison/agreement between FRCP and corrosion casting is beyond the scope of our study and would not add significant value to it. Additionally, corrosion casting is not a diagnostic technique, and its main role in the current study is to confirm the morphology of the biliary tract/pancreatic ducts. For example, corrosion casting allowed us to conclude that although in some MRCP and FRCP studies the hepatic ducts or the pancreatic ducts were not visible, they were still present.

Reviewer Counter: Reviewer comments are in an effort to improve the manuscript. I see in response to all of the reviewers the authors have failed to address the issues that the reviewers had (and potential readers will have). In this study you are using corrosion casting as a way of validating the FRCP data but have failed to provide compelling evidence of their agreement. That is a significant limitation in the study. This can be included as a limitation, but my impression is that the “scope” of this study is of little value without that information.

Response: We fully agree that reviewer comments are an extremely valuable tool for improving manuscripts, and we sincerely thank Reviewer 1 for his/her clear comments. However, the main aim of the present study was “to investigate the feasibility of MRCP in visualizing the biliary tract and pancreatic ducts in adult dogs with and without disorders of these structures. Another aim was to explore whether diameters for bile ducts, GB, and pancreatic ducts measured in MRCP agree with those of fluoroscopic retrograde cholangiopancreatography (FRCP, a modified ERCP technique for cadavers) and correlate with those of corrosion casting”. To keep the study consistent with its aims and taking into account that Reviewers 2 and 3 have already agreed with the present study design, we respectfully disagree with adding comparison of FRCP with corrosion casting to the Methods and Results of our study. The diameter measurements of each method are presented in Tables 3 and 4 for each structure.

Point 11: Not enough information is provided in table 2. Include what breeds were represented in the mixed breeds and what was each animal’s body condition score? It is difficult to interpret the relevance of the weight information for especially the mixed breeds (as written) because we don’t know if they are large or small for their breed.

Response 11: We respectfully disagree with this suggestion. The objective of our study was to assess the overall feasibility of MRCP in dogs, and the weight range of the dogs sufficiently shows their size for this purpose. Based on the information in the patient recording system, it is not possible to have further details of breeds represented in mixed breeds or of body condition score of the animals.

Reviewer Counter: See reviewer counter to point 6. Disagreeing because your recording system did not have that information is not a strong argument. BCS can be determined at the time of necropsy. Animals of small size but heavy weight (increased BCS) may have smaller pancreatic duct than animals of the same weight with low BCS. That directly impact the results of this study. If that information is not available that is a limitation in the value of the data and should be added to the study limitations section.

Response: Added as suggested (Discussion, lines 384-385).

Point 12: Add comparisons/agreement between FRCP and corrosion casting and the associated p-values for each structure in Table 3.

Response 12: Please see the response to Point 10.

Reviewer Counter: See reviewer counter to point 10.

Response: See the response to reviewer counter to point 10.

Point 13: Add concordance between FRCP and corrosion casting and the associated p-values for each structure in Table 5.

Response 13: Please see the response to Point 10.

Reviewer Counter: See reviewer counter to point 10.

Response: See the response to reviewer counter to point 10.
